# OpenReview forum: "GEP: A GCG-Based method for extracting personally identifiable information from chatbots built on small language models"
_ICLR.cc/2026/Conference — Submitted to ICLR 2026_

### Official Review · Reviewer_AS9k · 2025-10-21

**Soundness:** 3
**Presentation:** 3
**Contribution:** 1
**Rating:** 2
**Confidence:** 5

**Summary:**

While Small language models (SLMs) have become increasingly popular due to less energy and time consumption during training
and inference, the personally identifiable information (PII) leakage of SLMs for downstream tasks has yet to be explored. This paper explores PII leakage of the chatbot based on SLM. To this end, it initially finetunes a new chatbot (referred to as ChatBioGPT) based on the backbone of BioGPT using medical datasets Alpaca and HealthCareMagic.
It then proposes GEP, which is a greedy coordinate gradient-based (GCG) method which is designed for PII extraction. The experiments show an increment of up to 60X more leakage compared with the previous template-based methods.

**Strengths:**

Privacy analyses studies in the literature mostly emphasize LLMs or non-conversational settings. This paper focuses on SLM chatbots in healthcare surfaces which is practically relevant risk where resource-constrained models are more feasible for deployment.
It further has a marginal novelty in terms of repurposing GCG for PII extraction.
Last but not the least, the reported ASR gains are considerable over template prompts for a certain SLM.

**Weaknesses:**

There is a major weakness in terms of comparing the proposed solution with a wide range of privacy leakage studies in medical domain.
The study mostly compares against template prompts and does not reproduce stronger extraction baselines such as divergence-based attacks/un-alignment to extract memorized text or subject-driven probing tools.
The inserted PII is mainly name/symptom pairs which are generated or summarized from public datasets. While this is a good starting point, it omits several other realistic PII forms including but not limited to addresses, phone numbers, multi-field records. As this is main focus of the paper, it leads to concerns regarding contributions.
Another concern is only considering one SLM, i.e., BioGPT. There are several other SLMs for medical domain or general purposes (that could also include medical) such as BiomedGPT, BioMistral, and  TinyLlama. There is a gap in identifying whether this solution can be extended to other SLMs or not.

**Questions:**

-Can you add comparative results against divergence/un-alignment attacks and subject-driven probing, using proposed SLM-chat setup?
-Can you evaluate on more PII types (such as phone/email/address/date of birth and  PIIs specifically applicable in medical domain) and report precision/recall for various scenarios?
-Can you conduct experiments on at least two other biomedical SLMs such as BioMistral?
-Can you test on non-medical SLMs to verify whether the results depend on medical domain or can be generalized?
-Can Sensitive Information be Deleted from LLMs?
-While the increased ASR is promising, can you add a brief discussion of potential future research to address this issue from defense side?

---

> ### Author Response · Authors · 2025-11-21
>
> We thank the reviewer AS9k for the thoughtful feedback and the reviewer’s recognition of the practical relevance of our SLM-focused study and the effectiveness of our GCG-based approach. Please find our answers (A) to the reviewer’s questions (Q) as follows.
>
> (W1&Q1) Baselines beyond template prompts
>
> A: Our goal is to evaluate PII leakage in SLM-based chatbots under realistic medical-domain usage. Many divergence-based or un-alignment attack baselines were designed primarily for safety jailbreaks on aligned LLMs (e.g., extracting harmful instructions) rather than privacy extraction. These methods rely on assumptions (e.g., alignment weaknesses, or domain-general generation) that do not transfer directly to SLM medical chat settings.
> We therefore follow the baselines used in prior PII-extraction studies (e.g., Lukas et al., 2023; Lehman et al., 2021; Huang et al., 2022; Chen et al., 2024), which share the same privacy-focused threat model. We clarify this rationale in the revised text.
>
> (W2 and Q2) Additional PII types
>
> A: Our study focuses on the patient–disease relationship because it represents one of the most sensitive PII forms in clinical SLM deployments, where the domain is typically narrow and information-dense. In such scenarios, PII often concentrates into a small number of clinically relevant fields.
> We note in the manuscript that GEP optimizes trigger likelihoods through gradients and is not tied to a specific PII type; extending to other identifiers (email, phone, address) may be examined in later studies.
>
> (W3 and Q3/Q4) Additional SLMs for evaluation
>
> A: We agree that cross-model evaluation strengthens the contribution. Due to hardware constraints, running very large biomedical SLMs (e.g., BioMistral, BiomedGPT) falls outside the scope of this study, which focuses on models in the 1B or smaller range. We decide to add experiments on OPT-350M, which is similar in scale to BioGPT. Results are now added in Table 2 and Section 4.4.
>
> (Q5) “Sensitive Information Can Be Deleted from LLMs?”
>
> A: Prior work on machine unlearning and selective forgetting is relevant and can be viewed as a defense direction. However, domain-specific SLMs require careful handling because naive removal approaches may also suppress necessary medical content. We now mention this in the discussion as a potential research avenue.
>
> (Q6) Future defense research
>
> A:  We expanded the final paragraph of Section 5 to briefly outline challenges and opportunities for domain-sensitive defenses. Full defense design is complementary too, but outside the scope of this leakage-focused study.
>
> We thank the reviewer again for the insightful suggestions.

---

### Official Review · Reviewer_AYSg · 2025-10-28

**Soundness:** 2
**Presentation:** 2
**Contribution:** 2
**Rating:** 2
**Confidence:** 3

**Summary:**

This paper studies the privacy leakage risks of small language model (SLM)-based chatbots, focusing on personally identifiable information (PII). The authors fine-tune BioGPT into a new chatbot, ChatBioGPT, for the medical domain, and find that prior template-based PII extraction techniques are ineffective. To address this, they propose GEP (Greedy Coordinate Gradient-based PII extraction), which adapts gradient-based jailbreak methods (e.g., GCG) for PII attacks.
GEP is tested under both template-based and free-style PII insertion settings, achieving up to 60× higher extraction rates than baselines, and still exposing 4.53% leakage even in non-template, realistic scenarios.

**Strengths:**

- Novel research questions
- Methodological Contribution:

**Weaknesses:**

- Lack of generalization or cross-model validation
- Lack of practical motivation
- Lack of the discussion of the defense methods

**Questions:**

While gradient-based optimization for prompt generation (GCG, AutoPrompt, etc.) is not new, applying it to PII extraction under SLM constraints is novel and meaningful. The authors propose the GEP, the optimization methods to extract the personally identificable information from the small language models. The experimental results demonstrate that the proposed methods can achieve great performance on the information extracting. However, I do have serveral concerns as following:

- Lack of generalization.
The authors only consider one small language model in the experimental settings. There is no evidence that GEP generalizes to other architectures, domains, or unseen data distributions, thus the claimed general applicability remains unproven.

- Lack of practical motivation
The authors fail to discuss the threat model of the proposed attacks. In real world, it is very hard to obtain the parameters of the fine-tuned models on sensitive datasets, which makes the motivation of the attacks unpractical. Moreover, the authors also assume that the attacker know the patients' name and searches for disease leakage. This is a very specific form of PII and may not generalize to unseen entity types (emails, addresses, etc.).

- Lack of the discussion of the defense methods
If it is the real threat to the communtiy, the authors are supposed to discuss the potential defense methods against the proposed attacks.

---

> ### Author Response · Authors · 2025-11-21
>
> We thank the reviewer AYSg for the valuable comments and for highlighting the novelty of our research questions and acknowledging the methodological contribution of adapting gradient-based optimization for PII extraction in SLM chatbots. Please find our answers (A) to the comments on weakness (W) and questions (Q) as follows.
>
> (W1 and Q1) Generalization across SLMs
>
> A: Our primary focus in this work is medical SLM-based chatbots, where BioGPT is a widely used backbone. We agree that cross-model evidence strengthens the claims. We are therefore adding experiments on OPT-350M following the same training and evaluation pipeline, and will include results in Table 2 and a short Section 4.4. Broader cross-architecture and cross-domain evaluation is a natural next step, but outside the immediate scope of this medical-domain study.
>
> Regarding other PII types, our experiments center on the patient–disease relation because this is one of the most sensitive identifiers in medical settings. As noted in Section 3.2, We view this as an initial exploration focused on the medically most sensitive PII type; other identifiers may be examined in later studies.
>
> (W2 and Q2) Practical motivation and threat model
>
> A: Our threat model does not assume access to fine-tuned model on sensitive datasets. Instead, as discussed in Section 1, we consider the common scenario where large web-scale or crowd-sourced datasets contain residual, imperfectly cleaned PII—a known issue in both general and medical corpora. For example, in HealthCareMagic we observed real patient names that remained despite reported anonymization. The risk arises not from sensitive datasets but from imperfectly sanitized public or semi-public training data.
>
> Regarding attacker knowledge: we assume only awareness of an individual’s name, which is consistent with prior extraction settings and represents a minimal and realistic assumption in clinical or consumer chatbot deployments. Other PII forms may be incorporated in future work as noted above.
>
> (W3 and Q3) Discussion of defenses
>
> A: We expanded Section 5 to briefly discuss defenses. Some mechanisms used in large general-purpose LLMs (e.g., generic safety policies) do not directly transfer to domain-specific medical chatbots, where guardrails can inadvertently suppress necessary clinical information. We also note (Section 5) a potential way to incorporate perplexity considerations into the GEP optimization itself to bypass the perplexity filter. A more thorough defense evaluation is important, but complementary to our current focus on establishing the existence of leakage.
>
> We appreciate the reviewer’s thoughtful feedback and believe the added clarifications strengthen the paper.

---

### Official Review · Reviewer_vz9c · 2025-10-29

**Soundness:** 3
**Presentation:** 3
**Contribution:** 2
**Rating:** 4
**Confidence:** 4

**Summary:**

The paper studies PII leakage in chatbots built on small language models (SLMs), a comparatively underexplored setting versus LLMs. The authors fine-tune a biomedical chatbot, ChatBioGPT, and then probe it for leakage.  ChatBioGPT is built by fine-tuning BioGPT with Alpaca for general instruction following and HealthCareMagic-100k for medical Q&A; evaluation uses BERTScore on iCliniq.

**Strengths:**

+ Prior leakage work emphasizes LLMs or generic PLMs; this paper targets chatbots on SLMs and argues practical deployment risk in medicine.

+ Moving beyond fixed templates, the free-style rewrite is closer to what scraped corpora look like, and the method still extracts PII (ASR up to 4.53% with beam).

**Weaknesses:**

- Only names to disease/symptom pairs are used; symptoms are even “summarized in three words … using ChatGPT,” which risks templatic artifacts and semantic drift. Consider including broader PII types (addresses, emails) and non-medical identifiers.

- All core results are on one 347M BioGPT-based chatbot in the medical domain, limiting generality across SLM families and non-medical chats. (Model size mentioned in Table 2 row labels.)

- Comparisons to prior extraction work are mostly template-query methods with different setups; no evaluation against modern defense stacks (perplexity filters, semantic anomaly detectors, instruction-following guardrails).

- The technical contribution is limited as the main GCG method is widely used.

**Questions:**

1. How would GEP perform with standard guardrails enabled end-to-end (safety policies, regex and PII detectors, perplexity filtering) rather than raw model decoding? Any preliminary results?

2. In free-style insertion, you drop the “disease or symptom” surrogate. Can you report ablation where the surrogate phrase is paraphrased or partially masked to test robustness?

3. Can you isolate the effect of the system prompt by re-running T&T/T&G without it (or with a neutral system prompt) to quantify the degradation you qualitatively describe?

4. Does GEP transfer to other SLMs (e.g., OPT-350M, MiniCPM, OpenELM) and to non-medical chatbots? Even a small-scale cross-model table would help.

---

> ### Author Response · Authors · 2025-11-21
>
> We thank the reviewer vz9c for the helpful comments and for acknowledging our focus on SLM chatbots in medical deployment settings and for highlighting the value of our free-style insertion experiments, which demonstrate robust PII leakage detection beyond fixed templates. Please find our answers (A) to the comments on weakness (W) and questions (Q) as follows.
>
> (W1) PII types and ChatGPT summarization
> A: Our study focuses specifically on medical PII leakage in SLM-based clinical chatbots. The name–disease relation is among the most sensitive identifiers in this domain, and we clarified this motivation in Section 3.2. To avoid semantic drift in the three-word summaries to a maximum extent, we required ChatGPT to use only terms already present in the original entry and verified this with random checks. Broader PII types are an important direction for future work, but outside the current medical scope.
>
> (W2) Generality beyond BioGPT
> A: We agree that cross-model evaluation is valuable. We are extending experiments to OPT-350M using the same pipeline and will include results in Table 2 and a short Section 4.4. We aim to keep the current study focused while enabling broader investigation later.
>
> (W3) Comparison with defense stacks
> A: Our goal is to show that template-based querying underestimates leakage in SLM chatbots and that GEP exposes substantially more risk. Defense evaluation is an important next step; we added discussion in Section 5 about why some LLM defenses (e.g., generic safety policies) may not directly apply to domain-specific medical chatbots.
>
> (W4) Technical novelty
> A: While GCG is known, our contribution is adapting it for PII extraction under both template and free-style insertion settings, which belongs to the category of privacy issue, where prior work has not applied it, as it concentrates on the security attack. The empirical findings demonstrate substantially higher leakage than standard methods.
>
> Q1 - Guardrails
> A: We have not yet run GEP under full guardrail pipelines. As discussed in Section 5, certain defenses (e.g., perplexity filters) could have potential impacts on the extraction results but there are also ways to bypass it, while generic safety filters may suppress medically relevant content.
>
> Q2 - Paraphrasing/Masking surrogate phrases
> A: In free-style insertion, no consistent surrogate phrase exists, so paraphrasing or masking it is not applicable. The unified trigger mechanism already tests generalization across varied syntactic forms regarding different diseases.
>
> Q3 - System prompt effect
> A: Because we target chatbot SLMs, system prompts are part of the intended interface. We therefore report qualitative observations, supported by prior work (e.g., Nasr et al., 2023), and clarify this in Section 4.2.
>
> Q4 - Transfer to other SLMs
> A: We are running experiments on OPT-350M and add the results in Table.2 and section 4.4. Broader SLM and non-medical transfer is feasible but beyond the immediate scope.
>
> We appreciate the reviewer’s suggestions and believe these clarifications strengthen the paper.

---

### Official Review · Reviewer_xFgQ · 2025-10-31

**Soundness:** 2
**Presentation:** 2
**Contribution:** 2
**Rating:** 6
**Confidence:** 3

**Summary:**

This study explores the PII leakage of chat models based on SLM. The authors develop ChatBioGPT based on BioGPT. Also, the authors propose GEP for PII extraction. Experimental results reflect the vulnerability of SLMs to privacy issues.

**Strengths:**

The manuscript is easy to read. The proposed PII leakage attack outperforms template-based methods.

**Weaknesses:**

-  Is the assumption that P(d \mid q, \mathcal{T}, s) and P(s \mid q, \mathcal{T}) share the same trend too strong?
- For free-style insertion, the attack method uses part of the data to learn unified trigger tokens. The main concern is how to justify that this method works well in practice. It is possible that the attacker may have no prior information. The experiment setup seems to be less realistic. Since the authors generate the PII data by themself, the training data used to learn the unified trigger tokens could be highly similar to the remaining data.

**Questions:**

See weakness.

---

> ### Author Response · Authors · 2025-11-21
>
> We thank the reviewer xFgQ for the thoughtful feedback and the reviewer’s recognition of the manuscript’s clarity and the improved performance of our PII-extraction method. Please find our answers (A) to the comments on weakness (W) and questions (Q) as follows.
>
> (W1) Assumption on P(d \mid q, \mathcal{T}, s) vs. P(s \mid q, \mathcal{T})
>
> A: We agree this assumption is an approximation. Our intention is not to assert a theoretical identity, but to introduce a practical surrogate likelihood because the attacker lacks prior information about d. In the template-based insertion setting, the phrase “disease or symptom” is consistently adjacent to the disease token, making it a reliable anchor for optimization. Empirically, maximizing P(s \mid q, \mathcal{T}) substantially improves recovery of d, enabling 40–60× higher extraction compared with template-only baselines. We have clarified this motivation and empirical support in Section 3.3.2.
>
> (W2) Realism of unified trigger learning under free-style insertion
>
> A: We clarify that our setup aims to approximate a realistic adversarial scenario while keeping assumptions minimal:
> (1)The underlying texts come from HealthCareMagic-100k, a genuine patient–doctor dataset with naturally diverse phrasing.
> (2)Free-style insertion modifies only the subject (first → third person) and grammatical agreement; the structure and linguistic variability of the original conversations remain intact.
> (3)The 1k modified entries are randomly split into disjoint training/validation subsets. The unified triggers are learned only from the training subset and evaluated on unseen validation entries, ensuring non-overlap of phrasing.
>
> We have revised Sections 3.2.2 and 3.3.2 to articulate these points more clearly. We appreciate the reviewer’s comments and believe the clarification strengthens the paper.

---

### Meta-Review · Area_Chair_1ucA · 2026-01-07

**Summary:**

The paper studies leakage in chatbots built on small language models, the authors use a new biomedical chatbot (ChatBioGPT), based on the backbone of BioGPT, which they probe for leakage.

While the paper addresses an important issues, which was studied extensively for large models, but less so for smaller models, the execution did not convince the reviewer of the paper's merit at its current version.

Authors are encouraged to incorporate the several suggested improvements for their upcoming versions.

**Reviewer Concerns:**

reviewer AS9k concern on comparison with other privacy leakage studies remain valid despite updates in the paper

concerns (reviewers AYSg and AS9k) on discussing defence mechanisms were addressed in section 5

**Reviewer Scores:**

AS9k and AYSg could have maybe changed their 2 to 4

---

### Decision · Program_Chairs · 2026-01-26

Reject